# Bridging between material properties of proteins and the underlying molecular interactions

Guang Song[1,2]*

**1** Department of Computer Science, Iowa State University, Ames, IA, United States of America, **2** Program of Bioinformatics and Computational Biology, Iowa State University, Ames, IA, United States of America

* gsong@iastate.edu

## Abstract

In this work, we develop a novel method that bridges between material properties of proteins, particularly the modulus of elasticity, and the underlying molecular interactions. To this end, we employ both an all-atom normal mode analysis (NMA) model with the CHARMM force field and an elastic solid model for proteins and protein interfaces. And the "bridge" between the two models is a common physical property predictable by both models: the magnitude of thermal vibrations. This connection allows one to calibrate the Young's moduli of proteins and protein interface regions. We find that the Young's moduli of proteins are in the range of a few Gpa to 10 Gpa, while the Young's moduli of the interface regions are several times smaller. The work is significant as it represents the first attempt to systematically compute the elastic moduli of proteins from molecular interactions.

## 1 Introduction

Solid is one of the fundamental states of matter. Atoms in solids are packed tightly and kept together by physical interactions, such as ionic bonds (as in sodium chloride), covalent bonds (as in diamond), metalic bonds (as in metals), hydrogen bonds (as in ice), or van der Waals interactions (as in organic compounds) [1]. Solids held together mostly by van der Waals interactions are called van der Waals solids.

The type of force that keeps a solid together determines its material properties, such as elasticity. Solids held together by metalic, ionic, or covalent bonds appear stiff and have a much higher elastic moduli than solids held together mostly by non-bonded interactions such as proteins [2].

The elasticity of proteins has been studied in three primary ways, namely, protein fibers or fibrils [3], protein crystals [4], and protein capsids [5].

Among protein fibers, some exhibit high extensibility and have a very low Young's modulus, on the order of only 1–10 MPa. These fibers usually contain unstructured regions or high mobility motifs, undergo $\alpha$-helix to $\beta$-strand transitions, or even unfold under external forces, and consequently they appear very soft (1-10 Mpa) [3]. Other protein fibers are significantly different as they "form highly regular, nearly crystalline arrangement of monomer units

**Data Availability Statement:** Data are available at: https://github.com/gsongISU/sigmaESMrelease.

**Funding:** The author(s) received no specific funding for this work.

**Competing interests:** The authors have declared that no competing interests exist.

without regions that can extend, change secondary structure or unfold" [3] and thus appear much stiffer. Experimental studies showed these fibers, such as actin [6], tubulin, collagen [2], and keratin [7], fibrin [3], $\beta$-lactoglobulin fibrils [8], had an elastic modulus in the range of a few Gpa. The Young's moduli of these fiber proteins were obtained through direct measurement of force under stretching, mostly using atomic force spectroscopy (AFM) [8, 9].

The mechanical properties of proteins were studied also using protein crystals, mostly the crystals of hen egg-white lysozyme (HEWL), a model protein. Table 1 summarizes the techniques used, proteins studied, and results obtained. It lists also results from the compressibility studies done in solvent [10] and on a crystal structure solved at high pressure (1,000 bar) [11]. The results show a large variation, from less than 1 Gpa to over 10 Gpa. Compressibility study by Gavish et al. [10] showed that under most conditions, the adiabatic compressibilities of lysozyme, hemoglobin, and myoglobin in solution were on the order of 10-20% of the compressibility of water, or 5-7 $\times 10^{-11}$ $m^2/N$. Consequently, the bulk moduli of these proteins, which are the inverse of their compressibilities, are over 10 Gpa. (The bulk modulus or volume modulus should not be confused with Young's modulus or tensile modulus, though they tend to have similar values for most solids.) Kundrot and Richards [11] found that the isothermal compressibility of lysozyme proteins was 4.7 $10^{-3}$ kbar$^{-1}$, or 20 Gpa in bulk modulus. On the other hand, the Young's moduli obtained from protein crystals using techniques such as vibration, indentation, ultrasound, or Brillouin scattering are much smaller, from 0.3 Gpa to 5.5 Gpa (see Table 1). How to understand this discrepancy? It is possible that the larger values obtained from compressibility studies represent the elastic modulus of proteins themselves, while the elastic moduli of protein crystals probably represent the moduli of both proteins and protein interface regions, as well as other noncrystalline elements in the crystal cell such as intracrystalline liquid [4]. This would reconcile the seemingly conflicting results on elastic moduli of proteins in the literature.

The Young's moduli of proteins measured experimentally as reviewed above depend on a number of factors, including conditions of the samples (such as pH in solvent [10] or water content in crystals: moistened or dried [15], etc.), and the frequency range at which a measurement was carried out [17], which varies from static to kilohertz [12–14], to ultrasonic in the megahertz range [16, 17] and Brillouin light scattering method in the gigahertz range [18]. The elastic modulus of a protein is also temperature dependent. Morozov and Gevorkian analyzed that the mechanical properties of protein crystals at different temperatures and found that proteins became significantly more rigid below glass transition temperature, when the surface layer of proteins and their bound water became immobilized [4]. In this work, we also will look into the influence of temperature on a protein's elastic modulus.

The elastic modulus of proteins was studied also through protein capsids by nano-indentation using atomic force microscopy (AFM). Nano-indentation allows one to measure the stiffness of viral capsids. The elastic modulus of the capsid material can then be deduced from the

**Table 1. A summary of studies on the elasticity of proteins using protein crystals (top half), or compressibility measurements (bottom half).**

| Techniques | Proteins | Young's modulus | References |
|---|---|---|---|
| vibrating reed | lysozyme | 0.3-1.5 Gpa | [12–14] |
| indentation | lysozyme | 0.49-4.2 Gpa | [15] |
| ultrasound | lysozyme | 4.87-5.5 Gpa | [16, 17] |
| brillouin scattering | lysozyme | 5.5 Gpa | [18] |
| | | bulk modulus | |
| ultrasound in solution | lysozyme, Mb, Hb | >10 Gpa | [10] |
| X-ray at high pressure | lysozyme | >20 Gpa | [11] |

measured stiffness. However, in order to do so, in most cases a thin-shell model was assumed and thin-shell elasticity theory applied. The limitation of the thin shell model is that it neglects the molecular structure of the capsid and assumes a homogeneous material property throughout, as well as an idealized spherical geometry and uniform thickness [5]. Thick shell models combined with finite element analysis were also tried [19] and the effect of non-uniform geometry was investigated [20]. These work all represent a top-down approach since experimental determined stiffness values were used in all cases to fit the underlying elastic moduli of capsids. A review article in 2012 by Mateu [5] summarized the nano-indentation results of a dozen viral capsids: the stiffness of the viral capsids and the derived Young's moduli. The review showed the Young's moduli of capsid proteins varied over a span of more than one order of magnitude, from 0.14 GPa for CCMV to 1-3 GPa for MVM and Φ29 [5]. The range of variation in Young's moduli seen in capsids is consistent with what is seen in protein crystals (Table 1, top half) but is much lower overall than what is obtained from compressibility studies (Table 1, bottom half), A possible explanation is that elastic moduli thus obtained represent the average moduli of both proteins and protein interface regions, while the elastic moduli obtained from compressibility studies represent those of individual proteins.

The Young's moduli of globular and membrane proteins were also estimated theoretically or computationally [2, 6, 21, 22]. By approximating proteins as solid materials with sheets that interacted through van der Waals interactions, Howard estimated that the Young's modulus of proteins to be around 4 Gpa (see Appendix 3.1 of [2]). In the case of F-actin, Bathe applied axial stretching of his elastic solid model for actin to match experimental stretching stiffness data [6] and found that the effective Young's modulus of actin was 2.7 Gpa [21].

**Contribution of this work**. In this work, we develop a novel bottom-up approach for computing the elastic moduli of proteins from the underlying molecular interactions. Our approach utilizes both normal mode analysis (NMA) [23–25], a well established technique for studying the fluctuating dynamics of macromolecules, and elastic solid models (ESM) [26] developed more recently that treat macromolecules as elastic solids with material properties such as Young's modulus. Since both of these models can predict the magnitude of thermal vibrations of macromolecules, this commonality is used as a bridge to link material properties modeled in ESM [26] with molecular interactions used in NMA. To the best of our knowledge, This work represents the first attempt to determine the Young's moduli of proteins and protein interface regions separately and systematically from all-atom molecular interactions. We find the Young's modulus of proteins can be as high as 10 Gpa, while the Young's modulus of protein interface regions is several times smaller. Our work reconciles the high modulus values found through compressibility studies and low values found in protein crystals or capsids. The large span of variations of elastic moduli at interface regions provides an explanation also for the similar extent of variations seen in protein crystals and protein capsids.

## 2 Methods

### 2.1 $\sigma ESM$, a molecular surface-based elastic solid model

Recently, we presented a novel elastic solid model called $\alpha ESM$[26] for macromolecules based on alpha shape [27]. The model has a parameter alpha which was chosen empirically.

Here we present a new extension of $\alpha ESM$ whose major improvement over $\alpha ESM$ is that the alpha value is now tuned individually for each protein so that the final elastic solid model of the protein has the same volume as its solvent excluded volume, such as that computed by the MSMS algorithm [28]. Another significant difference of the present ESM model from $\alpha ESM$ is that in representing the solid, not only the atomic coordinates of the protein are used, which is the case for $\alpha ESM$, but also selected points on the protein's molecular surface [21].

```
 1 function [K,M,shp] = sigmaESM(xyz, verts, faces, volume, mass, E)
 2 % simgaESM: ESM based on molecular surface
 3 % Author: Guang Song
 4 % K, M: stiffness and mass matrices of the elastic solid
 5 % xyz: atomic coordinates of the input structure
 6 % verts, faces, volume: vertices and faces of MSMS surface and its enclosed volume
 7 % mass: mass of atoms of the input structure
 8 % E: Young's modulus
 9 if nargin<6 E = 1.0; end
10 init_alpha = 3.6; % 2.1 atom radius + 1.5 (probe radius)
11 shp0 = alphaShape(xyz, init_alpha);
12 [pface] = boundaryFacets(shp0);
13 m = size(pface,1); % m: # of faces on the initial alpha shape
14 % simplify the molecular surface to have m faces
15 [rF, rV] = reducepatch(faces, verts, m);
16 xyz = [xyz; rV]; % append reduced surface vertices rV to xyz
17 % binary search for alpha so that the alpha shape has the given volume
18 alpha = findAlpha(xyz, volume);
19 [K, M, shp] = alphaESM(xyz, alpha, mass, E);
```

**Fig 1. The algorithmic flow of *σESM* as expressed in a MATLAB script.** A complete copy of the above MATLAB script is available at https://github.com/gsongISU/sigmaESMrelease. The scripts for computing stiffness matrix and mass matrix used in *alphaESM* are available at MATLAB file exchange (https://www.mathworks.com/matlabcentral/fileexchange/27826-fast-fem-assembly-nodal-elements), kindly contributed by Anjam and Valdman [30].

Consequently, the final solid model has a similar surface to the molecular surface of the protein. In this aspect the model is similar to Bathe's elastic solid model [21]. We name this new variant of ESM *σESM*, since $\sigma$ is a symbol commonly used for surface area or surface tension [29].

The procedural flow of *σESM* is given in Fig 1. The script takes as input *xyz*, which contains the coordinates of all the heavy atoms of a given protein structure that is available in its PDB file, and *verts*, *faces*, and *volume*, all three of which are outputs of the MSMS software [28] and represent respectively the vertices and faces of the solvent excluded surface mesh and its enclosed volume (line 1). The program first computes an initial alpha shape of the protein alone (line 11), which is used to set the number of faces that should be on the surface mesh (lines 12-13). It then simplifies the surface mesh produced by MSMS using MATLAB script *reducepatch* (line 15). The vertices on the reduced molecular surface are then appended to the coordinates of the atoms in the protein (line 16). Lastly, an alpha value is selected (line 18) so that the final alpha shape has the same volume as the the volume computed from MSMS [28]. Once the alpha value is determined, the script used for *αESM*[26] can be adopted to compute the stiffness matrix **K** and mass matrix **M** (line 19), from which eigenmodes and eigen frequencies are computed (Fig 1 in Ref. [26]).

Fig 2 shows the *αESM* model of a protein (pdb-id: 1aqb) in red. The molecular surface, or solvent excluded surface, computed from the MSMS software [28] is shown in blue as a triangular surface mesh. The total volume of the *αESM* solid model (shown in red) is 16,600 Å$^3$, while the solvent excluded volume is 23,720 Å$^3$. Fig 2(B) shows the *σESM* model in cyan, which has an extra layer over the *αESM* model and resembles closely the molecular surface in

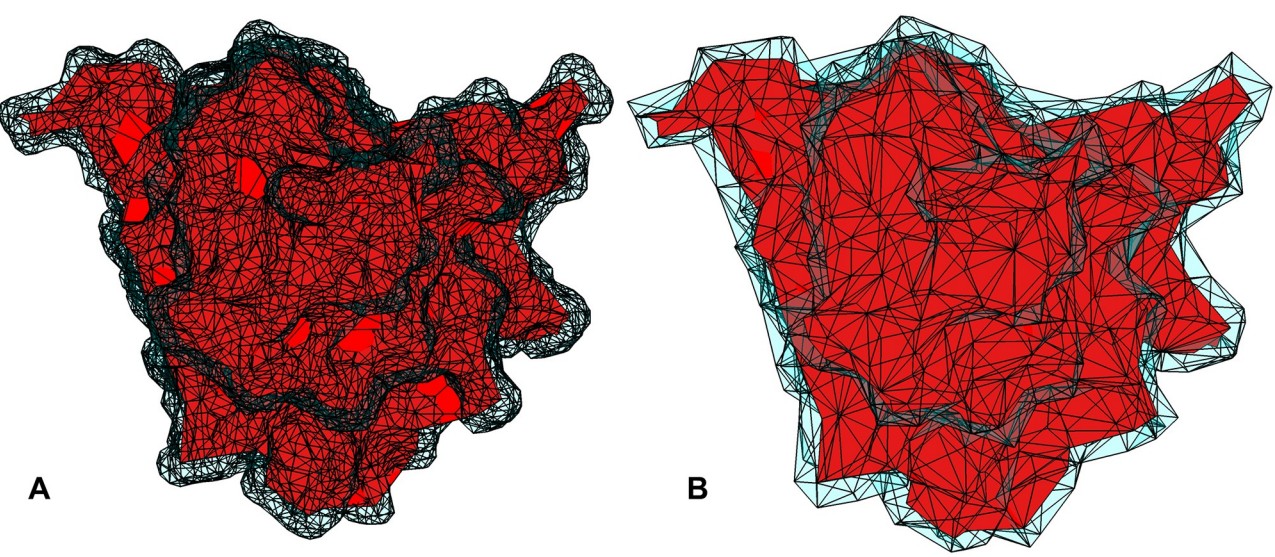

**Fig 2. The *αESM* model [26] (in red) using an alpha value of 3.6 Å.** The solvent excluded surface computed from MSMS [28] is shown in blue as a triangle mesh. (B) The *σESM* model: the difference between it and *αESM* (in red) is shown in cyan.

Fig 2(A). Specifically, the volume of *σESM* model is 23,770 Å$^3$, nearly the same as the protein's solvent excluded volume.

In summary, *σESM* is a more accurate representation than *αESM* in describing both molecular surface and and its enclosing volume.

**The magnitude of thermal vibrations by *σESM*.**   Given a protein structure and its molecular surface mesh (which can be generated for instance by the MSMS software [28]), an application of the *σESM* script given in Fig 1 produces a stiffness (**K**) and a mass (**M**) matrix. The (global) stiffness matrix **K** is a 3*M*-by-3*M* matrix, where *M* is the number of nodes in the model, built with the element stiffness matrices of all the elements (tetrahedral in the current work). Each element stiffness matrix is a 12x12 matrix that can be written as:

$$\mathbf{k} = [\mathbf{B}]^T [\mathbf{D}][\mathbf{B}] V,\tag{1}$$

where **B** and **D** are standard matrices available in textbooks on finite element method [31] that depend solely on the coordinates of the four nodes of the tetrahedral element, and *V* is the volume of the tetrahedron. The unit of **K** is kcal/mol/Å$^2$. For the mass matrix **M** used in finite element method, one has the option of a consistent mass matrix or a lump-sum mass matrix [31]. In this work, the lump-sum mass matrix is used, which is the same as the mass matrix used in NMA and is diagonal. (The consistent mass matrix, on the other hand, is not a diagonal matrix.) According to Ref. [26], the mean-square thermal vibrations of atom *i* by ESM is:

$$\langle \Delta \mathbf{R}_i^2 \rangle_{\mathrm{ESM}} = \frac{k_B T}{E} trace\left( \left[ \mathbf{M}^{-\frac{1}{2}} \mathbf{K}_m^{-1} \mathbf{M}^{-\frac{1}{2}} \right]_{ii} \right),\tag{2}$$

where the subscript *ii* represents the i$^{th}$ 3-by-3 diagonal block and $\mathbf{K}_m$ is the mass-weighted stiffness matrix, i.e., $\mathbf{K}_m = \mathbf{M}^{-1/2} \mathbf{K} \mathbf{M}^{-1/2}$. *E* is the Young's modulus of the protein solid.

The magnitude of thermal vibrations of a whole protein is defined as the *mass-weighted average* of $\langle \Delta \mathbf{R}_i^2 \rangle_{\mathrm{ESM}}$ over all the nodes [32], i.e.,

$$\langle \Delta \mathbf{R}^2 \rangle_{\mathrm{ESM}} = \frac{\sum_i m_i \langle \Delta \mathbf{R}_i^2 \rangle_{\mathrm{ESM}}}{\sum_i m_i} = \frac{k_B T \, trace(\mathbf{K}_m^{-1})}{E \sum_i m_i},\tag{3}$$

where $i$ is the index of the nodes that include both protein atoms and added molecular surface nodes. $m_i$ are the mass of node $i$. Each surface node is given a mass of 1 atomic unit.

## 2.2 The sbNMA model

The sbNMA model developed by Na and Song [33] is an all-atom NMA model based on the CHARMM force field [34]. It was designed to maintain the high accuracy of classical NMA (cNMA) using all-atom force fields. By circumventing the cumbersome step of energy minimization, it can be applied directly to experimental structures. Extensive studies have demonstrated its accuracy [32, 33, 35–37]. The entire sbNMA code is publicly available at https://github.com/htna/sbNMA-Matlab. Given a protein structure, an application of the sbNMA code (particularly the sbNMA_PSF.m script available at the above link) produces a Hessian matrix $\mathbf{H}$ and a mass matrix $\mathbf{M}$ that is diagonal (the sbNMA_PSF script actually produces the diagonal of the mass matrix as a vector, which can be easily converted to a mass matrix when needed). The Hessian matrix $\mathbf{H}$ is a 3N-by-3N matrix containing the second partial derivatives of the potential energy function with respect to the (x,y,z) coordinates of the $N$ atoms in a protein. The unit of the Hessian matrix is kcal/mol/Å$^2$.

Once the Hessian matrix $\mathbf{H}$ and the mass matrix $\mathbf{M}$ are obtained, the mean square fluctuation of atom $i$ is [33]:

$$\langle \Delta \mathbf{R}_i^2 \rangle_{\text{NMA}} = k_B T trace\left( \left[ \mathbf{M}^{-\frac{1}{2}} \mathbf{H}_m^{-1} \mathbf{M}^{-\frac{1}{2}} \right]_{ii} \right), \tag{4}$$

Thus,

$$\langle \Delta \mathbf{R}^2 \rangle_{\text{NMA}} = \frac{\sum_i m_i \langle \Delta \mathbf{R}_i^2 \rangle_{\text{NMA}}}{\sum_i m_i} = \frac{k_B T trace(\mathbf{H}_m^{-1})}{\sum_i m_i} \tag{5}$$

As will be shown later, the magnitude of thermal vibrations as computed from sbNMA in Eq (5) can be used to calibrate the Young's moduli of proteins.

## 2.3 B-factors, static disorder, and glass transition

The Debye–Waller factor (DWF), or B-factor as it is called in protein X-ray crystallography, is a factor used to describe for each atom the degree to which electron density spreads out. The spread around the mean position of each atom $i$ is generally modeled as a Gaussian function and the magnitude of the spread is characterized by $\langle u_i^2 \rangle$, the mean-square displacement from the mean position. The isotropic Debye-Waller factor, or B-factor, is related to $\langle u_i^2 \rangle$ as:

$$\langle u_i^2 \rangle = \frac{3}{8\pi^2} B_i. \tag{6}$$

The magnitude of mean-square displacement of a whole protein is defined as the *mass-weighted* averages of $\langle u_i^2 \rangle$ over all the atoms [32], i.e.,

$$\langle u^2 \rangle = \frac{\sum_i m_i \langle u_i^2 \rangle}{\sum_i m_i}, \tag{7}$$

where $i$ is the atom index.

## 2.4 Determine the Young's moduli of proteins based on molecular interactions

sbNMA is an accurate NMA model for predicting the magnitude of thermal vibrations. However, like all NMA models and elastic network models, one major limitation of sbNMA is that it assumes harmonicity. Thus the predicted magnitude by sbNMA represents only a *lower bound* of the actual magnitude of fluctuations of proteins that may contain also anharmonic motions, especially at above $T_g$, the temperature of glass transition [38].

The magnitude of thermal vibrations can be obtained also experimentally. X-ray crystallography that is widely used in protein structure determination produces such magnitude information in atomic displacement parameters that are commonly known as B-factors. Since B-factors contain also a significant amount of static disorder [32], the magnitude of mean-square displacement obtained from B-factors thus represents an *upper bound* of the actual magnitude of fluctuations of proteins.

A possible third alternative for obtaining the magnitude of thermal fluctuations of proteins is molecular dynamics (MD) simulation [39, 40], which is able to take into account anharmonic motions and is unaffected by static disorder. However, it suffers from the problem of insufficient sampling of the conformation space. Consequently, the actual magnitude of thermal fluctuations is difficult to obtain using MD.

The Young's modulus represents a material property of a solid. By representing a protein using both a molecular model (as in NMA) and an elastic solid model as in *σESM*, we can link material properties with molecular interactions. Particularly, we can deduce a protein's Young's modulus by requiring the magnitude of thermal vibrations predicted by *σESM* in Eq (3) to be the same as the magnitude obtained from sbNMA in Eq (5) or the magnitude of mean-square displacement from B-factors in Eq (7). Since the magnitudes obtained from sbNMA and B-factors represent the lower and upper bounds of the actual magnitude respectively, the Young's moduli thus deduced represent the upper and lower bounds of the actual Young's moduli. Specifically, by requiring $\langle \Delta \mathbf{R}^2 \rangle_{\text{ESM}} = \langle \Delta \mathbf{R}^2 \rangle_{\text{NMA}}$ (Eqs (3) and (5)), we have:

$$E_{\text{upper}} = \frac{trace(\mathbf{K}_m^{-1})/m_{tot}^{\text{ESM}}}{trace(\mathbf{H}_m^{-1})/m_{tot}^{\text{NMA}}}, \tag{8}$$

where $m_{tot}^{\text{ESM}}$ and $m_{tot}^{\text{NMA}}$ are the total masses in *σESM* and NMA, respectively. And similarly, by requiring $\langle \Delta \mathbf{R}^2 \rangle_{\text{ESM}} = \langle u^2 \rangle$ (Eqs (3) and (7)), we have:

$$E_{\text{lower}} = \frac{trace(\mathbf{K}_m^{-1})/m_{tot}^{\text{ESM}}}{(\sum_i m_i \langle u_i^2 \rangle)/m_{tot}^{Bfac}}, \tag{9}$$

where $m_{tot}^{Bfac}$ is the total mass of all the atoms with B-factors, which usually are the heavy atoms.

## 2.5 Quaternary structures based on crystal contacts

For each protein chain $p$ in the dataset, we determine the all the other chains in its unit cell based on the space group and symmetry matrices given in its PDB file. Additionally we find all the neighboring cells to the current cell in all directions: altogether 27-1 = 26 neighboring cells are considered. Of all the chains in these cells, we identify the one that is the closest to $p$ and denote this chain as $q$. $p$ and $q$ form a crystal contact and a quaternary structure. (In the case when the selected $q$ has a very small contact surface with $p$, a different chain $q$ is used instead.) The quaternary structures are saved in PDB files as xxxx2.pdb, where xxxx is the pdb-id, and are available at https://github.com/gsongISU/sigmaESMrelease, under folder *pdbDataset*. In

the following, we explain how to compute the Young's modulus of the interface region of a quaternary structure.

## 2.6 $\sigma ESM$ of the quaternary structures

$\sigma ESM$ is readily applicable to the aforementioned quaternary structures in the same fashion as to regular tertiary structures, and as a result, an elastic solid model is produced for each quaternary structure (see Fig 3 of structure 2pwa2.pdb). The nodes in the model consists of heavy atoms of two protein chains (regions in red) in the quaternary structure and the surface nodes (in cyan) given by the MSMS software [28]. The elements in the solid model are tetrahedra generated by alpha shape [27] (see Fig 1). Given the elastic solid model of a quaternary structure, the interface region, which is in blue in Fig 3, is identified as follows.

First, all the tetrahedral elements formed by atoms from chain $p$ or surface nodes but not from chain $q$ are identified. Let all the nodes of these elements be $nodes1$. $nodes1$ consists of all the atoms of chain $p$ and some surface nodes, but no atoms from chain $q$. Similarly, compute $nodes2$ for the second chain $q$. Next, remove any common surface nodes shared by $nodes1$ and $nodes2$ from them. As a result, the intersection of $nodes1$ and $nodes2$ is an empty set. Finally, identify all the tetrahedral elements formed solely by $nodes1$ or $nodes2$. The remaining elements are identified as the interface region (in blue).

## 2.7 Computing the Young's modulus of the interface region

Next, we compute the Young's modulus of the interface region of the quaternary structure. This allows us to assess how "soft" or "firm" the region is. Again, our method to determine the material property of the interface region is to link it with the underlying molecular interactions using $\sigma ESM$ and sbNMA, by requiring the two models to give the same magnitude of thermal vibrations.

Fig 3(B) shows the molecular structure model of the same quaternary structure after protonation using psfgen from VMD [42]. We apply sbNMA to compute the magnitude of thermal vibration using Eq (5). To focus on the interface region and to speed up the computations, we set the two proteins as rigid, i.e., the internal degrees of freedom within each protein chain is left out, by employing a projection matrix as was done in the RTB model [43]. The magnitude of thermal vibration of the whole structure (Fig 3(B)) is thus dictated solely by inter-protein interactions. Similarly, when applying $\sigma ESM$ to the solid model in Fig 3(A), the two protein regions, which are in red, are set as rigid and only the interface region (in blue) is flexible. By requiring the two models to give the same magnitude of thermal vibration, we can determine the Young's modulus of the interface region, similar to what is done in Eq (8). The entire program for computing the Young's modulus of the interface region is given at https://github.com/gsongISU/sigmaESMrelease. The program uses $\sigma ESM$ and sbNMA and a matrix projection module. All of them are available at the above website.

It is worth noting that only non-bonded interactions (possibly also disulfide bonds) are present at the interface region. Therefore it is expected that the Young's modulus of the interface region should be smaller than that of a protein itself, since the latter contains also bonded interactions. Also, quaternary structures constructed from crystal contacts as described above, especially after protonation, may contain steric clashing in the interface region. For atom pairs that are too close to each other (i.e., closer than their equilibrium separation distance which is the sum of their van der Waals radii), we use their equilibrium separation distances as their separation distances when computing the potential energy at the interface.

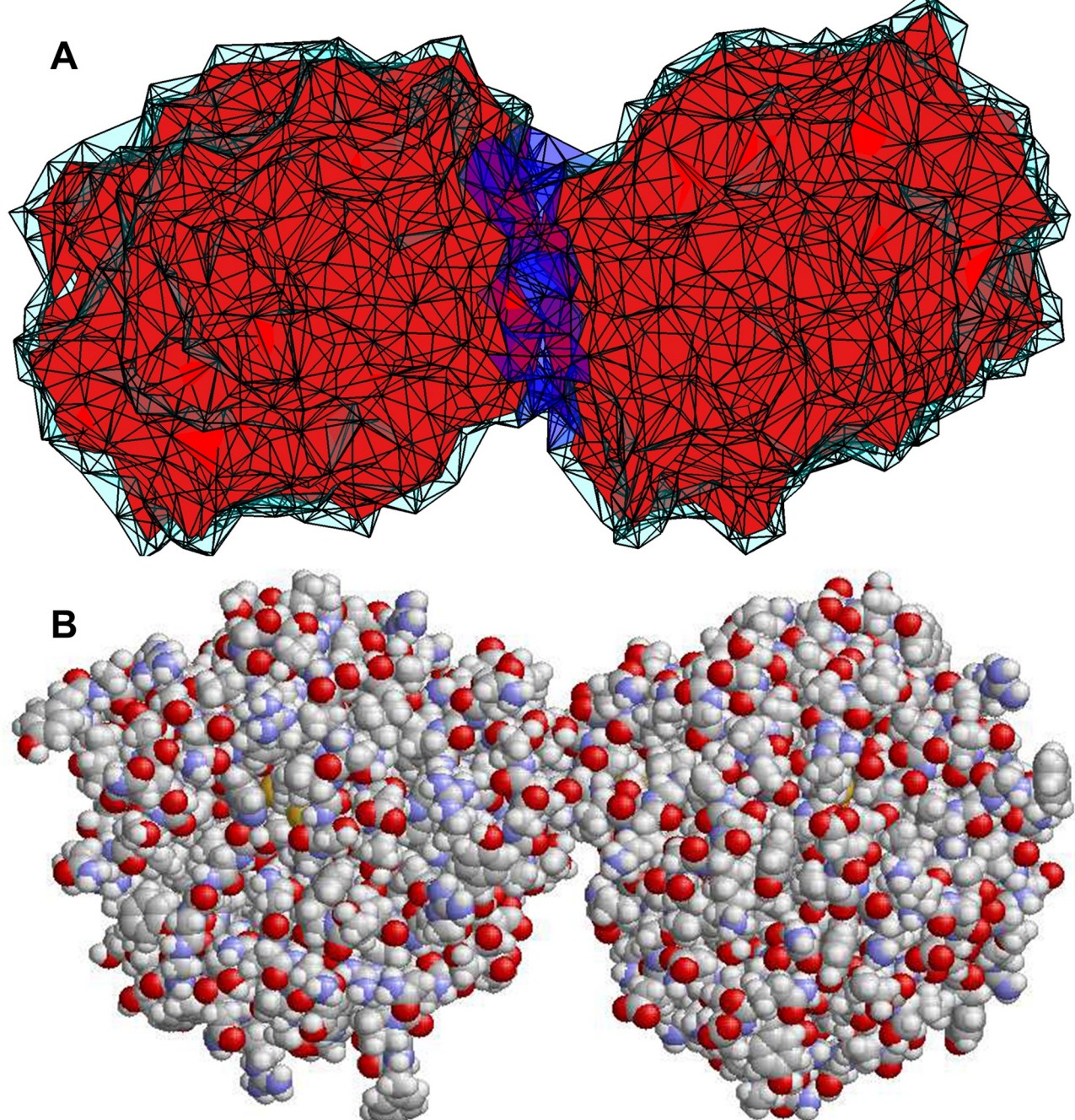

**Fig 3. Models of a quaternary structure.** (A) An elastic solid model of a quaternary structure (2pwa2.pdb). The pdb quaternary structure is available at https://github.com/gsongISU/sigmaESMrelease, under folder *pdbDataset*. (B) the all-atomic molecular model of the same structure. The image is generated by Rasmol [41].

## 3 Results

### 3.1 The choice of protein dataset

In our recent work [32], we have shown that the percentage of thermal vibrations in B-factors is small in most proteins but is significantly large (50% or higher) for structures determined at

both high resolution (1.2 Å or higher) and high temperature (that is, room temperature). Unfortunately such structures are rare in the PDB, as most structures in the PDB [44] are determined at cryogenic temperature to reduce the radiation damage [45].

Based on this premise, we select a group of 18 structures determined at high resolution (1.1 Å or high) and at room temperature. This set of structures are taken from a list of 1522 structures used in our previous work [32] that was originally generated by PDB_SELECT [46, 47].

## 3.2 Upper bound and lower bound of thermal fluctuations in proteins

There are a few universal properties of globular proteins that are intriguing. First, it was established that the vibrational spectra of globular proteins, once properly normalized, follow a universal curve [35, 48]. It was shown later that this universality in vibrational spectrum held true also for protein capsids [36]. Second, it was shown in one of our recent work that a direct corollary of the universality in vibrational spectrum among globular proteins is that the magnitudes of their thermal vibrations are nearly universal, having a narrow distribution that peaks at 0.093 Å$^2$ at 100 K [32]. In obtaining this value, a harmonic potential is assumed. The harmonicity assumption is reasonable for thermal vibrations at or below 100 K. At higher temperature, particularly at above the glass transition temperature of proteins which is about 200-220 K, proteins undergo "glass transition" and become much more flexible [38]. As a result, the magnitude of mean-square displacements grows super-linearly. Therefore, the magnitude of thermal vibrations of globular proteins at 300 K is at least three times as high as 0.093 Å$^2$, or 0.279 Å$^2$. In other words, the magnitude of thermal vibrations computed from sbNMA (which assumes harmonicity) represents a lower bound of the actual magnitude at room temperature.

On the other hand, the magnitudes of thermal fluctuations of globular proteins are available in crystallographic B-factors, though in which unfortunately a large amount of static disorder co-resides. Because of static disorder, B-factors do not represent the actual magnitudes of thermal fluctuations but rather an upper bound.

The aforementioned upper and lower bounds define a range of the actual magnitudes of thermal vibrations of globular proteins. The range, though not as ideal as a single definite value, gives us a good sense of the magnitude.

Fig 4 shows the upper and lower bounds of the magnitudes of thermal fluctuations of the 18 structures in the dataset. For three out of the 18 structures, the lower bound obtained from the sbNMA computation is slightly higher than the upper bound obtained from B-factors. This could be due to the imperfection of sbNMA or uncertainties in B-factors, or both. However, for most proteins, the over trend is that the upper bound is about 1.6-1.7 times higher than the lower bound. The mean lower and upper bounds of the proteins shown in Fig 4 are 0.30 and 0.51 Å$^2$, respectively.

## 3.3 Glass transition

It is known that proteins undergo glass transition at $T_g$, the glass transition temperature, which is about 200-220 K for proteins [38]. At above $T_g$, proteins become much more flexible. Past work indicated that "the glass transition in hydrated samples is located in the surface layer of proteins and related to the (im)mobilization of the protein groups and strongly bound water." [4]. As a result, the magnitude of mean-square displacements grows super-linearly with temperature.

Fig 5 presents the magnitudes of thermal vibrations as computed from sbNMA [32, 33] of the 18 proteins in the dataset, as well as their mean-square displacements from crystallographic B-factors. The blue line represents the linear growth of mean-square displacements as a

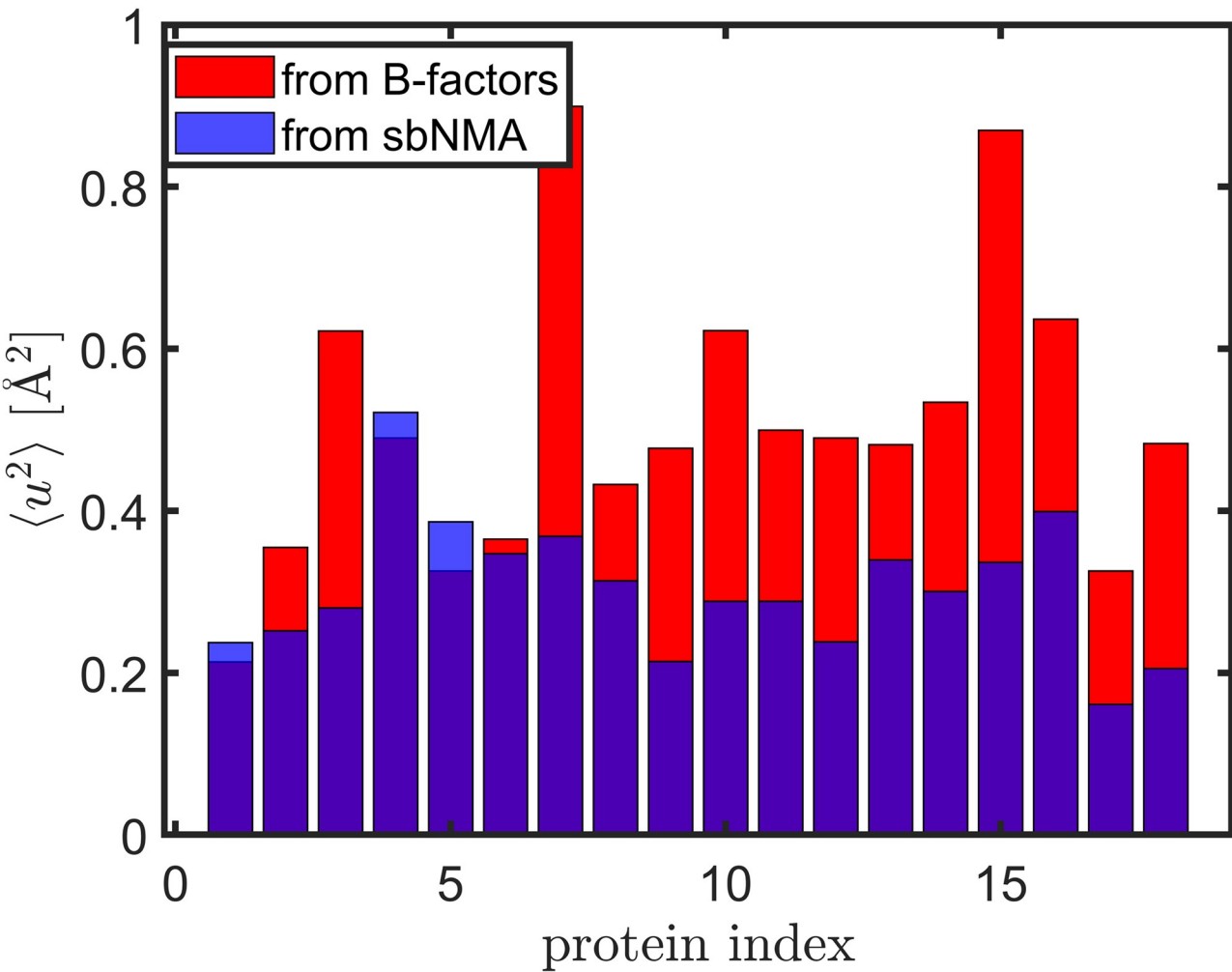

**Fig 4. The magnitudes of thermal fluctuations obtained from B-factors (in red) and from sbNMA computations (in blue), which form the upper and lower bounds of the magnitudes of thermal fluctuations of the 18 structures in the dataset, listed by their sizes in the ascending order.** For three out of the 18 structures, the lower bound is slightly higher than the upper bound.

function of temperature should there be no glass transition. The red line in Fig 5 represents a fitting to $\langle u^2 \rangle$ (of B-factors). A clear picture of glass transition emerges.

### 3.4 The range of Young's modulus

Fig 6 shows the range of Young's moduli computed from $\sigma ESM$ and calibrated with sbNMA or B-factors. The median Young's modulus when fitting to B-factors is 6.03 Gpa, and 10.6 Gpa when fitting to sbNMA. sbNMA does not consider glass transition but only harmonic motions. Thus, the Young's moduli calibrated with sbNMA can be thought of as the Young's moduli of proteins at temperatures below $T_g$, while the Young's moduli fit to B-factors can be thought of as Young's moduli at the room temperature.

### 3.5 Estimating the Young's modulus of the interface region using sbNMA and CHARMM force field

The interface region is kept together mostly by non-bonded electrostatic and van der Waals interactions. Electrostatic interactions are difficult to include in normal mode analysis as they

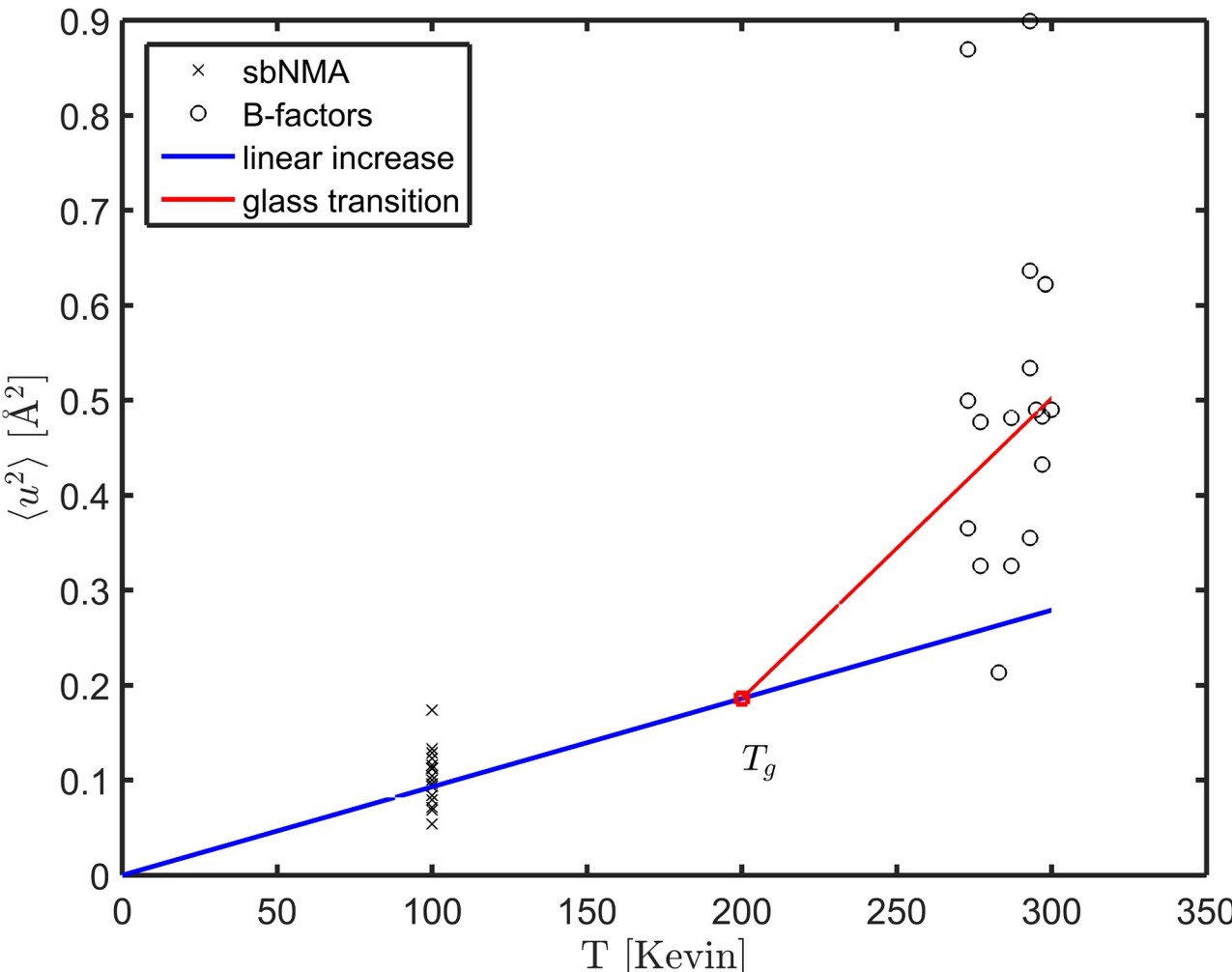

**Fig 5. The magnitude of thermal fluctuations as a function of temperature.** $T_g$ denotes the temperature of glass transition that is 200-220K [38]. The magnitudes computed from sbNMA at 100K are marked by black crosses, while the magnitudes obtained from B-factors by black circles. The blue line represents the universal trend of the magnitude of harmonic vibrations by sbNMA with CHARMM force field, having a nearly universal value around 0.093 Å$^2$ at 100 K for all proteins [32], and 0.279 Å$^2$ at 300 K. At above $T_g$, anharmonic motions kick in and the magnitude of total thermal fluctuations increases superlinearly, as marked by the red line that is fitted to the black circles.

introduce negative spring constants [33]. Our past studies showed that van der Waals interactions provided a greater contribution than the electrostatic interactions (Fig 1 in Ref. [33]) and a model without electrostatic interactions could still maintain most of the accuracy of the classical NMA [33]. For these reasons, our current model for the interactions at the interface includes only the van der Waals interactions. It is planned that a future release of the model should improve this deficiency. The van der Waals potential employed in the CHARMM force field is a 6-12 Leonard-Jones potential that takes the following form:

$$V_{\text{vdW}} = \epsilon \left( \left( \frac{r_0}{r} \right)^{12} - 2 \left( \frac{r_0}{r} \right)^{6} \right). \tag{10}$$

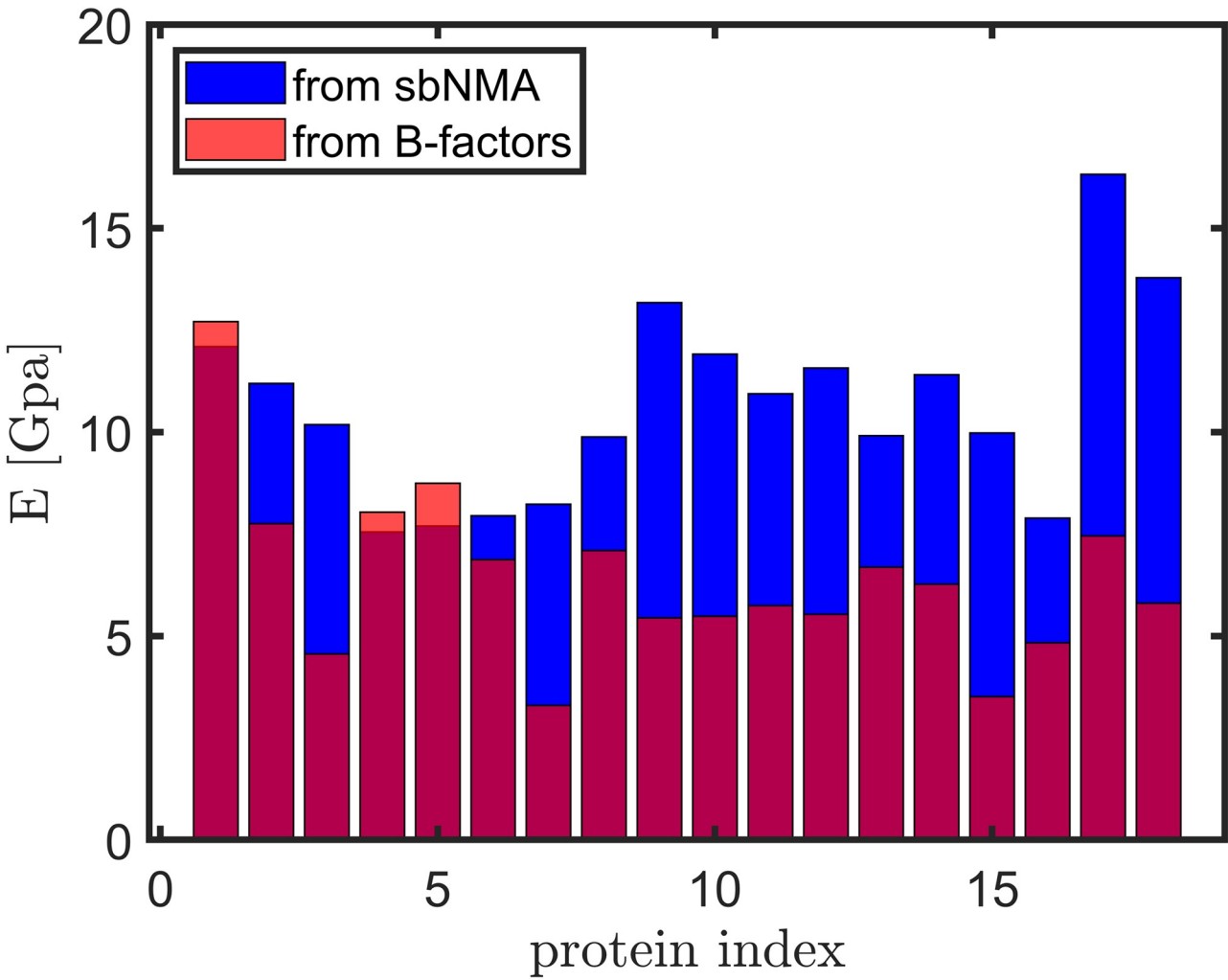

**Fig 6. A bar plot of the Young's moduli ($E$) calibrated with sbNMA (in blue) and those with crystallographic B-factors (in red), for the 18 proteins in the dataset (listed by their sizes in the ascending order).** The blue bars can be interpreted also the Young's moduli at temperatures below $T_g$, while the red bars Young's moduli at the room temperature.

The spring constant due to the van der Waals potential is:

$$k_{\text{vdW}} = \frac{\partial^2 V_{\text{vdW}}}{\partial r^2} = \frac{12\epsilon}{r^2}\left(13\left(\frac{r_0}{r}\right)^{12} - 7\left(\frac{r_0}{r}\right)^6\right) = \frac{72\epsilon}{r_0^2}, \tag{11}$$

the last step of which assumes $r = r_0$.

Fig 7 shows the histogram distribution of the van der Waals parameter $\epsilon$ of the 50 CHARMM atom types in par_all36_prot.prm and the histogram distribution of $k_{vdW}(r_0)$, which falls mostly in the range of 0.1 to 1 Kcal/mol/Å$^2$. It is thus clear that $k_{vdW}$, the spring constant due to van der Waals interactions, is much weaker than the spring constants for covalent bonds that are in the order of several hundreds Kcal/mol/Å$^2$.

Since the equilibrium distance between a pair of atoms are in the range of 2.5 to 4.5 Å, The Young's modulus of the interface region, $E_{inter}$, is estimated to be on the order of $\frac{k_{vdW}}{r_0}$, or 0.02 to 0.4 Kcal/mol/Å$^3$ (or 0.1 to 3 Gpa, the converting factor between the two units is 6.9). This is

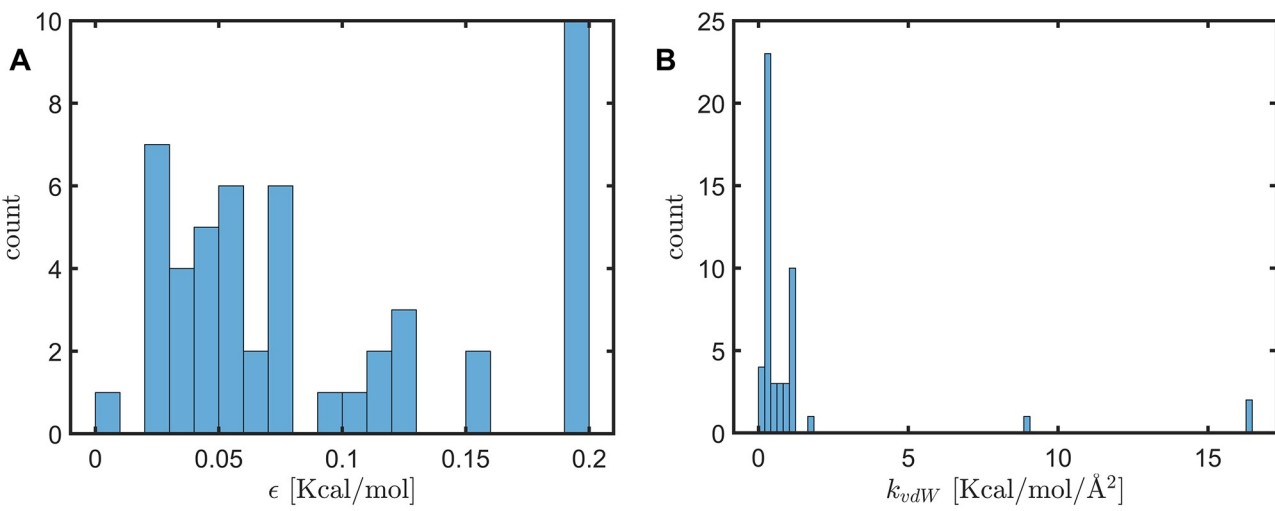

**Fig 7. Histogram distributions of (A) the van der Waals parameter $\epsilon$ used in CHARMM and (B) of $k_{vdW}(r_0)$, the spring constant of van der Waal interactions.**

2 to 3 orders smaller than that of covalent solids or ionic solids, which is expected since spring constants derived from van der Waals interactions are 2 to 3 orders smaller than those of covalent bonds or ionic bonds.

## 3.6 Young's moduli of the protein interface regions

Using the approach described in the Methods, we first generate quaternary structures of all the 18 proteins in the dataset. $\sigma ESM$ (see Methods section for details) is then applied to compute the Young's moduli of the interface regions (the MATLAB scripts used for this computation are available at https://github.com/gsongISU/sigmaESMrelease). The results are given in Table 2. It is seen that while the Young's moduli of the proteins are a few Gpa, the Young's moduli of interface regions are about an order smaller. Such a significant difference in stiffness between proteins and protein interfaces must be due to the underlying chain connectivity of proteins and the associated bonded interactions. Proteins can be considered as van der Waals solids with "steel enforcement" [49]. Also listed in Table 2 are the buried surface areas of interface regions of the quaternary structures. The buried surface area is the difference between the total surface area of individual proteins and that of the complex, both of which are computed using MSMS [28].

The above 18 quaternary structures are artificially generated based on crystal contacts. An insightful reviewer commented that interfaces thus identified tend to be fortuitous and recommended that real protein-protein interfaces existing naturally should also be used and a comparison be made. For this reason, we selected another 18 high resolution structures (1.0 Å or higher) from the PDB that are in the form of homodimers in the asymmetric unit. Their pdb-ids and sizes are given in Table 3, as well as the buried surface areas and the Young's moduli of the interface regions.

To compare the Young's moduli of protein interfaces of the naturally existing dimers (Table 3) with those created by crystal contacts (Table 2), we show in Fig 8 a scatter plot between the buried surface areas and the Young's moduli of the interface regions, for both natural interfaces (red dots) and artificial interfaces created by crystal contacts (blue crosses). First, a significant correlation is seen between the buried surface areas and the Young's moduli (correlation coefficient: 0.77). Secondly, The natural interfaces (red dots) have a much higher

**Table 2. The lower and upper bounds of Young's moduli of the 18 proteins and the Young's moduli of the interface regions of the corresponding quaternary structures artificially generated based on crystal contacts.**

| pdb-id | residue # | buried area [Å²] | $E_{inter}$ | $E_{intra}$ (lower bound) | $E_{intra}$ (upper bound) |
|---|---|---|---|---|---|
| 1p9g | 41 | 75 | 0.09 | 12.71 | 12.10 |
| 1rb9 | 52 | 263 | 0.58 | 7.76 | 11.19 |
| 1iro | 53 | 388 | 0.56 | 4.56 | 10.19 |
| 2igd | 61 | 171 | 0.40 | 8.04 | 7.56 |
| 1aho | 64 | 493 | 1.17 | 8.75 | 7.70 |
| 5tog | 75 | 541 | 0.56 | 6.88 | 7.95 |
| 1ctj | 89 | 352 | 0.96 | 3.29 | 8.23 |
| 1lwb | 122 | 251 | 0.58 | 7.10 | 9.88 |
| 1c7k | 132 | 81 | 0.36 | 5.45 | 13.17 |
| 3r87 | 132 | 837 | 0.83 | 5.48 | 11.91 |
| 6gz8 | 133 | 293 | 1.13 | 5.75 | 10.95 |
| 5vg0 | 142 | 181 | 0.23 | 5.53 | 11.57 |
| 4b9g | 146 | 264 | 1.02 | 6.69 | 9.91 |
| 4ann | 176 | 527 | 0.57 | 6.27 | 11.41 |
| 4qa8 | 210 | 401 | 0.91 | 3.51 | 9.99 |
| 6h40 | 220 | 183 | 0.57 | 4.83 | 7.89 |
| 2pwa | 279 | 203 | 0.44 | 7.45 | 16.32 |
| 6gy5 | 285 | 103 | 0.15 | 5.80 | 13.79 |
| median | 132 | 263 | 0.57 | 6.03 | 10.57 |

The lower bounds are obtained by fitting σESM results to B-factors, while the higher bounds are obtained by fitting the σESM results to sbNMA. Also listed in the table are the sizes of the 18 proteins and the buried surface areas of the interface regions. The unit for all Young's moduli is Gpa.

**Table 3. The buried surface areas and the Young's moduli ($E_{inter}$) of the natural interfaces found in a list of 18 high resolution structures from the PDB that are in the form of homodimers in the asymmetric unit.**

| pdb-id | residue # | buried surface area [Å²] | $E_{inter}$ [Gpa] |
|---|---|---|---|
| 4ynh | 58 | 1,964 | 2.06 |
| 3rq9 | 84 | 900 | 1.60 |
| 4nds | 94 | 1,279 | 2.26 |
| 2nmz | 99 | 2,381 | 3.12 |
| 4unu | 109 | 597 | 0.53 |
| 6j64 | 115 | 2,695 | 2.69 |
| 2xr4 | 116 | 1,987 | 1.89 |
| 4egu | 118 | 2,256 | 2.02 |
| 2gud | 121 | 2,997 | 3.50 |
| 4axo | 137 | 2,615 | 2.09 |
| 5nld | 138 | 1,058 | 1.15 |
| 5idb | 142 | 1,390 | 1.89 |
| 2wyt | 153 | 729 | 1.71 |
| 4a7v | 153 | 704 | 1.50 |
| 5sy4 | 195 | 1,802 | 2.14 |
| 6rk0 | 214 | 5,313 | 3.77 |
| 3noq | 229 | 3,327 | 2.17 |
| 4ypo | 325 | 7,924 | 2.50 |
| median | 129 | 1,976 | 2.08 |

The buried surface areas and the Young's moduli of these natural interfaces are distinctly higher than those of the artificial interfaces seen in Table 2.

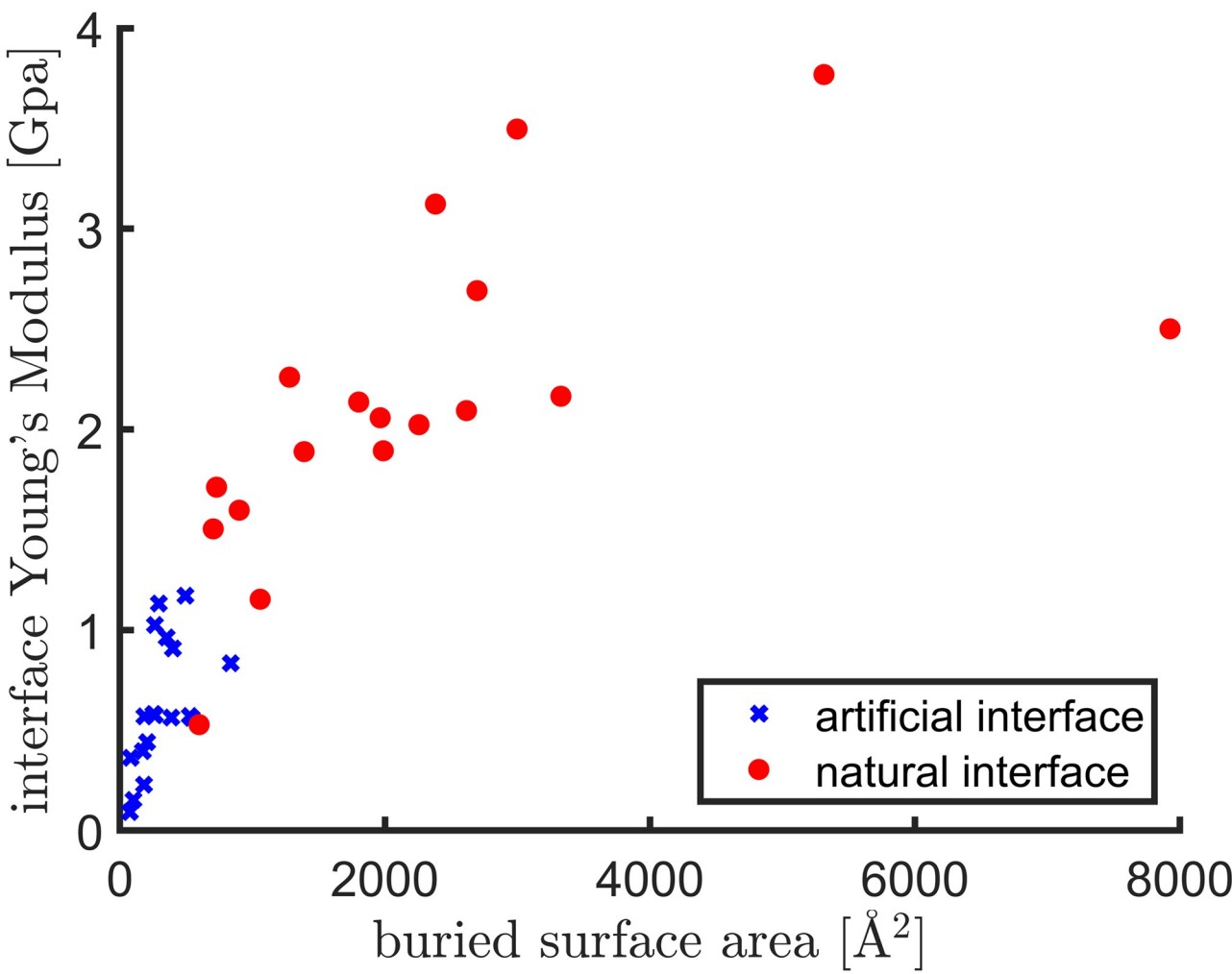

**Fig 8. A scatter plot between the buried surface areas and the Young's moduli of the interface regions, for both artificial (blue crosses, data points from Table 2) and natural (red solid circles, data points from Table 3) interfaces.**

buried surface areas, though the two sets of proteins have similar sizes, and are much stiffer, having a median Young's modulus of about 2.0 Gpa that is about 4 times as high as that of artificial interfaces. Thirdly, even for the natural interfaces, the Young's moduli are still a few times lower than those of proteins. Since a clear distinction is seen between artificial interfaces and natural interfaces in Fig 8, a potential application of the present study is to use the interface's Young's modulus proposed in this work to assess the quality of a predicted protein interface.

## 4 Discussion

In this work, we have studied the elastic moduli of proteins and protein interfaces using a bottom-up approach. The material properties of solids are determined by the underlying physical interactions. For proteins and protein interfaces in particular, their material properties such as elasticity are dictated by the underlying molecular interactions.

The innovation of this work is the development of a novel method that can bridge between material properties at the macroscopic level and molecular interactions at the microscopic

level. To achieve this, we employ both an all-atom NMA model (here sbNMA [33]) and an elastic solid model ($\sigma ESM$) for proteins and protein interfaces. And the "bridge" between the two models is a common physical property predictable by both models: the magnitude of thermal vibrations (see Eq (8)), i.e.,

$$\langle \Delta \mathbf{R}^2 \rangle_{\mathrm{ESM}} = \langle \Delta \mathbf{R}^2 \rangle_{\mathrm{NMA}} \tag{12}$$

This connection allows us to determine the Young's moduli of proteins and protein interfaces using molecular interactions.

A similar bridge can be built between $\sigma ESM$ and crystallographic B-factors (Eq 9) as well. A significant benefit of doing both is that the two constraints are complementary to each other and together they provide both an upper and lower bound for the Young's moluli of proteins.

Atoms at the interface region interact primarily through non-bonded interactions. One key realization is that the spring constant due to the van der Waals interaction, or $k_{vdW}$, is small and is mostly in the range of 0.1-1 kcal/mol/Å$^2$ (Fig 7). Since the separation distances $r_0$ between pairs of atoms at the interface region should be about the sum of their van der Waals radii, one may estimate the order of Young's modulus at the interface region by simply taking the ratio of the two, which comes to be about 0.1 to 3 Gpa. Our actual computations are in agreement with this estimation. Covalent solids such as diamond have a Young's modulus that is hundreds of Gpa or even over a thousand Gpa, which is 2-3 orders higher than that of proteins or protein interfaces. The difference has a simple physical explanation: the spring constants for covalent bonds (as in CHARMM force field) are 2-3 orders higher than the van der Waals spring constants while the separation distances between interacting atoms are similar. In a nutshell, the low Young's modulus at protein interface is directly due to the weak non-bonded interactions.

Another interesting finding is that, compared with protein interfaces, the Young's modulus of proteins is several times higher. After all, proteins are chains of amino acids linked together by covalent bonds. The covalent chain serves as a "steel enforcement" [49] that stiffens a protein solid. The abundance of secondary structures in proteins provides additional enforcement through hydrogen bonds. As a result, the internal of a protein is several times stiffer than protein interface.

As part of future work, we plan to apply the method to study the elastic properties of protein capsids. The stiffness of viral capsids has been measured for a large number of viruses using AFM technique [5]. The stiffness varies greatly among the viral capsids, with that of some capsids being an order of magnitude higher than that of others [5]. The physical cause of this large degree of variations in stiffness is not known. As mentioned in the Introduction, existing work, using with either the thin shell or thick shell models, all represents a top-down approach that uses experimental determined stiffness values to fit the underlying elastic moduli of capsids and were not able to provide a physical explanation for the observed stiffness. Our present approach has the potential to take into the account not only the structural details of protein capsids (instead of a shell with uniform spherical geometry and thickness), but also the striking difference in elastic properties between protein subunits and protein interface regions, and consequently may be used to *predict* capsid stiffness and provide an explanation for the observed differences in stiffness among viral capsids.

## Author Contributions

**Conceptualization:** Guang Song.

**Data curation:** Guang Song.

**Formal analysis:** Guang Song.

**Investigation:** Guang Song.

**Methodology:** Guang Song.

**Resources:** Guang Song.

**Software:** Guang Song.

**Supervision:** Guang Song.

**Validation:** Guang Song.

**Writing – original draft:** Guang Song.

**Writing – review & editing:** Guang Song.

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
