## [Decision Letter · Decision Letter 0]

12 Mar 2021

PONE-D-21-02708

Bridging Between Material Properties Of Proteins And The Underlying Molecular Interactions

PLOS ONE

Dear Dr. Song,

Thank you for submitting your manuscript to PLOS ONE. After careful consideration, we feel that it has merit but does not fully meet PLOS ONE’s publication criteria as it currently stands. Therefore, we invite you to submit a revised version of the manuscript that addresses the all the points raised by both reviewers.

We look forward to receiving your revised manuscript.

Kind regards,

Claudio M. Soares, Ph.D

Academic Editor

PLOS ONE

Journal Requirements:

Reviewers' comments:

Reviewer's Responses to Questions

**Comments to the Author**

1. Is the manuscript technically sound, and do the data support the conclusions?

Reviewer #1: Partly

Reviewer #2: No

2. Has the statistical analysis been performed appropriately and rigorously? 

Reviewer #1: Yes

Reviewer #2: Yes

3. Have the authors made all data underlying the findings in their manuscript fully available?

Reviewer #1: Yes

Reviewer #2: Yes

4. Is the manuscript presented in an intelligible fashion and written in standard English?

Reviewer #1: Yes

Reviewer #2: Yes

5. Review Comments to the Author

Reviewer #1: This article by Song reports the fluctuations in the interfacial region between proteins, interpreted as a material property. The author draws an interesting analogy between the mechanical properties of fibrils and the interfacial mechanics of the proteins. The key point appears to be justified: the fluctuation of the interface between proteins is softer than the internal fluctuations of a protein because of the different interactions and degrees of freedom.

The key point missing from this paper is that the mechanical model is not truly required for looking at the fluctuations of proteins; they really are not isotropic materials. Discussion of this really must be done to make the point of the article.

My comments below mostly complain about the naive analysis of the van der waals interactions between proteins which I think needs to be modified.

Major comments:

In my opinion normal mode analysis does not really qualify as predicting thermal vibrations because it is using the thermal vibrations to compute the modes. Casting it in terms of an elastic material does not change this. What the model can do is predict the strain in response to stress. Alternatively, the author might cast it as the way of predicting the thermal vibrations of large fibers from a local property of the dimer (abstract).

Proteins aren’t mostly held together by van der waals forces. Here you must know that this could only refer to the fold of a protein. The atoms of a protein are connected by covalent bonds. The hydrophobic effect and hydrogen bonding are mostly at play. The hydrophobic effect is not the van der waals force (Section 1). “The interface region is kept together mostly by van der Waals interactions” (Section 3.5).

Model peptides and water near the interface region experience relatively strong Coulomb interactions with the protein and are repelled by the vdW approximation to the Pauli repulsion at close contact (the 1/r12 potential). I believe there would still be a fluctuating interface even in the complete absence of an attractive 1/r^6 potential. Therefore, the estimate is unjustified.

The fluctuating interface shown in Figure 3 is interesting but because it is a novel result and appears challenging to validate a more sophisticated estimate would be interesting to see. Is this interface the result of hydrophobic contacts? Can the fluctuations of the hydrophobic contact volume be explained in terms of the hydrophobic-effect-driven tension? This would seem to be a more fruitful explanation than purely in terms of vdW contacts.

Figure 6 is unacceptable. Considering protein index is not important (x-axis) and the values per-protein are listed in a table, I suggest a plot of sbNMA Young’s modulus vs. the B-factor young’s modulus. For example, when the crystallographic Young’s modulus (red) is greater than the sbNMA analysis (blue) the blue value is not shown. An XY/scatter plot would indicate the correlation of differences in protein Young’s modulus.

Minor:

“Along with liquid, gas, and plasma, solid is one of the four states of matter.” Wikipedia lists quite a few more such as Bose-Einstein condensates. There is no reason to make such obvious statements in a scientific article. I urge the author to be less didactic (Section 1).

The bulk modulus is not the Young’s modulus and the author confuses them in the introduction. The Young’s modulus refers to strain along a direction with the two other directions unconstrained. Water is a fluid and as such has a zero Young’s modulus (Section 1).

What is the rationale for using the mass-weighted averages as opposed to anything else? Mass has nothing to do with any elastic modulus; it is independent of mass-scaling (2.3).

Lessen et al, JCTC 2018 is another computational study attempting to compute the Young’s modulus: https://dx.doi.org/10.1021/acs.jctc.8b00377

Reviewer #2: This manuscript discusses calculations and results on the elasticity of proteins and their interfaces. The underlying ideas is to equate the results of calculations based on solid elastic models with those based on atomic vibrations either from atomistic MD simulations or crystallographic temperature factors. The issue of elasticity of proteins and their interfaces is an interesting one, and the basic mathematical idea is sound and well executed. There is however a very important problem with the paper with regard to how the author has chosen to think about protein interfaces, perhaps reflecting some unfamiliarity with how structural biologists think about protein quaternary structure vs incidental crystal contacts. Basically, the underlying idea of the paper is interesting and perhaps informative, but to be meaningful the study would have to be repeated with a different set of classifications of what is considered a protein interface.

Major problem:

The author takes crystal structures from the PDB and generates protein interfaces by examining additional chains that are in contact based on crystallographic operations. By itself, this is not an appropriate procedure for defining protein interfaces. The study doesn’t make the critical distinction between real protein-protein interfaces (i.e. those that have arisen in nature by evolution) and fortuitous contacts (often very small) that occur incidentally during crystallization. The former category is of substantial interest. The latter category might also be of interest in a more specialized context. But it is vital that the two things not be confused or conflated. There is a long literature on both subjects. There are also dedicated databased set up to help disentangle the two cases, which is not always easy. The PISA protein database is one. The PQS protein quaternary structure database is another. Just to be more explicit, one could go to such a resource and ask for a list of PDB structures where the protein molecule is believed to be a monomer in its native form (comprised of just one protein chain); for those cases, contacts present in the expanded crystal state would (presumably) represent fortuitous protein-protein crystal contacts. For analyzing natural protein-protein interfaces, one can query the database for proteins believed to exist as dimers in their natural biological forms. Then one can extract the dimeric forms directly from that curated database – obtaining the correct biological forms is not always trivial to do by expanding from the coordinates in the PDB as the author has attempted to do based on closest contact. Analysis of these cases would then represent evolved protein-protein interfaces. And of course quaternary structures higher than dimers could be considered, with added complexity. Once the author does this, then the results, which are likely to differ substantially, can be analyzed in separate categories of natural and incidental crystal interfaces. It is a virtual certainty that the cases of natural interfaces will not be as flexible as the whole set of protein-protein contacts (including incidental crystal contacts) which the author has presently defined.

Minor points:

1) At the bottom of page 8, the form and units of the stiffness matrix K should be given. E.g. is it the collection of second partial derivatives of the conformational energy (normalized by kT) with respect to atomic coordinates (or finite elements)? Similarly for the Hessian at the top of page 10.

2) The legend to Figure 4 should clarify that the two kinds of experimental results are being interpreted as upper and lower bounds.

3) The second paragraph on page 11 should introduce, at least parenthetically, that the accepted technical term for ‘B-factors’ is ‘atomic displacement parameters’.

4) Protein-protein interfaces are customarily discussed/plotted with respect to buried surface area rather than volumes.

6. PLOS authors have the option to publish the peer review history of their article (what does this mean?). If published, this will include your full peer review and any attached files.

Reviewer #1: No

Reviewer #2: No

---

## [Author Response · Author response to Decision Letter 0]

2 Apr 2021

see attached PDF: responsesToReviewers.pdf

---

## [Decision Letter · Decision Letter 1]

20 Apr 2021

Bridging Between Material Properties Of Proteins And The Underlying Molecular Interactions

PONE-D-21-02708R1

Dear Dr. Song,

We’re pleased to inform you that your manuscript has been judged scientifically suitable for publication and will be formally accepted for publication once it meets all outstanding technical requirements.

Kind regards,

Claudio M. Soares, Ph.D

Academic Editor

PLOS ONE

Additional Editor Comments (optional):

Reviewers' comments:

Reviewer's Responses to Questions

**Comments to the Author**

1. If the authors have adequately addressed your comments raised in a previous round of review and you feel that this manuscript is now acceptable for publication, you may indicate that here to bypass the “Comments to the Author” section, enter your conflict of interest statement in the “Confidential to Editor” section, and submit your "Accept" recommendation.

Reviewer #1: All comments have been addressed

Reviewer #2: All comments have been addressed

2. Is the manuscript technically sound, and do the data support the conclusions?

Reviewer #1: Yes

Reviewer #2: Yes

3. Has the statistical analysis been performed appropriately and rigorously? 

Reviewer #1: Yes

Reviewer #2: Yes

4. Have the authors made all data underlying the findings in their manuscript fully available?

Reviewer #1: Yes

Reviewer #2: Yes

5. Is the manuscript presented in an intelligible fashion and written in standard English?

Reviewer #1: Yes

Reviewer #2: Yes

6. Review Comments to the Author

Reviewer #1: Thank you for addressing my comments. I look forward to seeing electrostatics addressed in the model.

Reviewer #2: The revised version is much improved and ready for publication in its present form. The inclusion of natural interfaces now makes the paper useful.

7. PLOS authors have the option to publish the peer review history of their article (what does this mean?). If published, this will include your full peer review and any attached files.

Reviewer #1: No

Reviewer #2: No

---

## [Editor Report · Acceptance letter]

23 Apr 2021

PONE-D-21-02708R1 

Bridging Between Material Properties Of ProteinsAnd The Underlying Molecular Interactions 

Dear Dr. Song:

I'm pleased to inform you that your manuscript has been deemed suitable for publication in PLOS ONE. Congratulations! Your manuscript is now with our production department. 

Kind regards, 

on behalf of

Dr. Claudio M. Soares 

Academic Editor

PLOS ONE